# Myelinating Co-Culture as a Model to Study Anti-NMDAR Neurotoxicity

**DOI:** 10.3390/ijms24010248

**Published:** 2022-12-23

**Authors:** Mercedeh Farhat Sabet, Sumanta Barman, Mathias Beller, Sven G. Meuth, Nico Melzer, Orhan Aktas, Norbert Goebels, Tim Prozorovski

**Affiliations:** 1Department of Neurology, Medical Faculty, Heinrich-Heine-Universität Düsseldorf, 40225 Düsseldorf, Germany; 2Institut für Mathematische Modellierung Biologischer Systeme, Heinrich-Heine-Universität Düsseldorf, 40225 Düsseldorf, Germany

**Keywords:** anti-NMDAR, GluN1, myelinated cultures, NRHypo, degeneration

## Abstract

Anti-NMDA receptor (NMDAR) encephalitis is frequently associated with demyelinating disorders (e.g., multiple sclerosis (MS), neuromyelitis optica spectrum disorder (NMOSD), myelin oligodendrocyte glycoprotein-associated disease (MOGAD)) with regard to clinical presentation, neuropathological and cerebrospinal fluid findings. Indeed, autoantibodies (AABs) against the GluN1 (NR1) subunit of the NMDAR diminish glutamatergic transmission in both neurons and oligodendrocytes, leading to a state of NMDAR hypofunction. Considering the vital role of oligodendroglial NMDAR signaling in neuron-glia communication and, in particular, in tightly regulated trophic support to neurons, the influence of GluN1 targeting on the physiology of myelinated axon may be of importance. We applied a myelinating spinal cord cell culture model that contains all major CNS cell types, to evaluate the effects of a patient-derived GluN1-specific monoclonal antibody (SSM5) on neuronal and myelin integrity. A non-brain reactive (12D7) antibody was used as the corresponding isotype control. We show that in cultures at the late stage of myelination, prolonged treatment with SSM5, but not 12D7, leads to neuronal damage. This is characterized by neurite blebbing and fragmentation, and a reduction in the number of myelinated axons. However, this significant toxic effect of SSM5 was not observed in earlier cultures at the beginning of myelination. Anti-GluN1 AABs induce neurodegenerative changes and associated myelin loss in myelinated spinal cord cultures. These findings may point to the higher vulnerability of myelinated neurons towards interference in glutamatergic communication, and may refer to the disturbance of the NMDAR-mediated oligodendrocyte metabolic supply. Our work contributes to the understanding of the emerging association of NMDAR encephalitis with demyelinating disorders.

## 1. Introduction

Anti-N-methyl-D-aspartate receptor (NMDAR) encephalitis is a severe neuropsychiatric immunological disorder, which is diagnosed by the presence of circulating anti-NMDAR autoantibodies in the cerebral spinal fluid (CSF) [1,2,3]. The clinical symptoms developed by patients are thought to be due to substantial alterations of glutamatergic transmission, and include psychiatric and cognitive manifestations that progress to movement disorder, speech dysfunction, seizures, impaired consciousness, and autonomic instability. Often anti-NMDAR encephalitis is also associated with tumors, mostly ovarian teratoma [4]. Early immunotherapy and/or tumor removal leads to a substantial or even complete recovery in about 80% of patients [5]. Histopathological examination demonstrates extensive microgliosis and the presence of IgG deposits in the hippocampus, basal forebrain, basal ganglia, and spinal cord, including signatures of endoneural edema and Wallerian degeneration in affected neural tissue [6]. Approximately 5% of patients develop clinical and radiological evidence of a demyelinating disorder [7].

Autoantibodies to glutamate receptors with a proven pathological role include the immunoglobulins G (IgG) directed against ionotropic receptors AMPAR-GluR3, NMDAR-GluN1, -GluN2A, and -GluN2B, as well as AABs against metabotropic receptors (anti-mGluR1 and anti-mGluR5 glycoproteins) (reviewed in [8]). In contrast to other anti-NMDARs (e.g., anti-GluN2A or anti-GluN2B) eliciting excitotoxicity via receptor activation [8], crosslinking of NMDARs by autoantibodies to the extracellular domain of the obligatory GluN1 subunit affects glutamate signaling via mechanisms involving receptor internalization, decreased surface cluster density, and synaptic localization of the NMDARs [9,10]. Of note, compensatory synaptic plasticity mechanisms activated following the exposure to anti-GluN1 antibodies do not induce counterbalancing changes in the expression of the glutamate receptor; however, they cause a decreased inhibitory synapse density onto excitatory neurons [11]. The effect of human anti-GluN1 IgG on inhibitory and excitatory neurons demonstrates brain region-specific alterations. In contrast to hippocampal neurons, dysfunction of inhibitory neuron output by anti-GluN1 IgG triggers a hyperexcitable state in cultivated cortical neurons leading to network hyperactivity [12]. NMDAR hypofunction has been proposed to be a part of the pathophysiological mechanisms underlying anti-GluN1-driven anti-NMDAR encephalitis [13].

We have previously isolated an anti-NMDAR-GluN1 (SSM5) autoantibody from intrathecal plasma cells of a patient with anti-NMDAR encephalitis and demonstrated in vivo its pathogenic relevance [14]. We showed that the effect of SSM5 is not only restricted to neurons, and may also affect other cell types in which glutamatergic transmission plays a role. For example, anti-GluN1 IgG affects the NMDAR-induced Ca^2+^ response and the glucose transporter 1 (GLUT1; SLC2A1) surface expression in myelinating oligodendrocytes [15]. This effect may strikingly influence trophic oligodendrocyte-neuron communication and energetic support of myelinated axons through the recently described glutamate-mediated signaling in axon-myelin synapses [16].

In the present study, we examine the effects of SSM5 and a corresponding isotype control, non-brain reactive monoclonal IgG1 (12D7) [14], particularly in myelinated neurons. We apply a robust model of long-term cultures derived from dissociated embryonic mouse spinal cord [17]. During cultivation, spinal cord cultures produce compactly myelinated internodes separated by nodes of Ranvier [18] and generate neurons that recapitulate spinal-like electrical activity [19], serving as a valuable tool for functional studies in myelinated, synapse-forming CNS neurons [20]. Besides neurons and oligodendrocytes that constitute the major cell population, the typical E12.5-E13.5 murine spinal cord-derived culture contains astrocytes and microglia that recapitulate innate CNS immune functions [21,22]. We show that myelinating spinal cord culture is a suitable model to investigate the effects of human anti-GluN1 antibodies and report that targeting GluN1 disturbs neuronal integrity and myelination, an effect that has been disregarded in other culture models. 

## 2. Results

### 2.1. Defining an Optimal Stage of Culture for Treatment

Mouse embryonic spinal cord cells were isolated from 12.5 days old embryos (E12.5) and maintained in culture, as described previously [17]. Cells were cultivated for 5 weeks, and progression of neuronal growth and myelination were examined. By day 14 in vitro (DIV), neurite lengthening, visualized by antibodies to neuronal differentiation marker neurofilament 200 kDa (NF200) [23,24], was completed (Figure 1). Upon differentiation, oligodendrocytes developed most of their processes aligned with axons, and were labelled by antibodies against myelin basic protein (MBP). Myelination starts between 3 and 4 weeks of culture (DIV 21–28) and peaks around DIV 35 (Figure 1b). Although not all oligodendrocytes become fully mature, about two-third of MBP-positive cells provided single or multiple myelin segments. The extent of neuronal insulation at the peak of myelination (DIV 35) was proportional to neurite density in individual wells (Figure 1c).

Next, we tested developing spinal cord cultures for sensitivity to NMDA in the presence of glycine (co-agonist; available in the culture medium at estimated concentration 0.2–0.25 mM). In line with previous data [25], overstimulation of spinal cord neurons with 100 µM NMDA starting at DIV 21, DIV 28 or DIV 35 lead to a massive loss of neurofilament (Appendix A) and widespread neuronal degeneration. Neuronal vulnerability to NMDA coincided with the loss of myelin (Figure 1b and Appendix A). 

Supplementation of cultures for one week with 1 µM thyroid hormone triiodothyronine (T3) that supports functional maturation and myelination during oligodendrocyte development [26], increased density of myelin segments in differentiating cultures maintained for DIV 21–35 (Appendix A). Notably, treatment with T3 starting after DIV 35 had no effect on myelination, indicating that oligodendrocyte maturation is mainly completed during the first 5 weeks in vitro (Figure 1b). Taken together, characterization of long-term cultures demonstrated that at DIV 21–35, differentiating neural cells develop functional NMDARs and are sensitive to cell growth-promoting mediators.

### 2.2. Immunoreactivity of SSM5 in Spinal Cord Cultures

Immunodetection with patient-derived anti-NMDAR-GluN1 antibody (SSM5) showed a dot-like immunoreactivity to neuronal processes starting around DIV 14–21 (Figure 2). SSM5 also co-localized with a fraction of chondroitin sulphate proteo-glycan (NG2)-positive cells (Appendix A) and myelin (MBP) aligned with axons (Figure 2). These results suggest that SSM5 affects glutamatergic transmission in neuronal and oligodendroglial cells derived from the embryonic spinal cord.

### 2.3. SSM5 Antibody Induces Neuronal Damage and Myelin Loss

Next, we studied the potential effect of SSM5 on neuronal survival and myelination. In order to avoid the negative effects of GluN1 targeting on neurite outgrowth and spine maturation of progenitor cells [27,28], we used 3- to 5-week-old cultures (DIV 21–35)–the period at which most neurons exhibited differentiated morphology (Figure 1a). 10 µg/mL SSM5, a concentration in culture media shown to be effective for the blocking of NMDA-evoked Ca2+ influx in neurons [14], were added to the myelinating cultures, and the effects were compared to isotype (12D7; 10 µg/mL) and non-treated control cultures. One-week treatment with SSM5 revealed a significant decrease in neurofilament content in late (DIV 28–42) cultures, while isotype control (12D7; 10 µg/mL) had no toxic effect (Figure 3). Loss of neurofilament immunoreactivity occurred progressively and reached highest values in cultures treated starting from DIV 35 (6% less of neurofilament at DIV 21–28; 18% at DIV 28–35; 23% at DIV 35–42 in comparison to the respective 12D7 controls), indicating that susceptibility to SSM5 toxicity is related to the stage of neural differentiation (Figure 3b). In parallel to the neurotoxic effect, SSM5 significantly lowered myelination in late cultures (7% less myelin segments at DIV 21–28; 18% at DIV 28–35; 21% at DIV 35–42 as compared to 12D7 group) (Figure 3c). We also found a significant difference in the percentage of nuclei exhibiting signs of pyknosis or karyorrhexis (DIV 21–28: 17% vs. 26% of abnormal nuclei; DIV 35–42: 19% vs. 28% of abnormal nuclei, compared to 12D7 group) (Appendix A).

To characterize the affected neurons, we analyzed neurite blebbing and fragmentation as markers of neurodegeneration during both spinal cord neuronal injury [29,30] and autoimmune antibody-mediated neurodegeneration [31,32]. Evaluation of neuronal morphology in cultures exposed to SSM5 revealed an increase in the number of injured neuronal processes (up to 8% at week DIV 21–28, 26% at DIV 28–35 and 33% at DIV 35–42; compared to 12D7 group) (Figure 4). Importantly, the severity of SSM5 neurotoxicity increased in late cultures characterized by higher proportion of myelinated axons (see Figure 1b). Thus, it is tempting to speculate that axonal myelination made them more vulnerable to neurotoxic stimulus. The number of injured neurons in cultures treated with 12D7 antibodies appeared similar to non-treated cultures.

We anticipated the myelin and neurite swellings induced by the SSM5. Analysis of damaged processes that remain individual myelinated segments indicates that changes in the myelin sheath structure (presumably myelin blisters [33]) occur at the site of axonal injury characterized by structural blebbing (Figure 5). Demyelinated axonal lesions frequently flank normally myelinated fragments (without axonal blebbing), hinting that the local area of damage serves as a platform for further myelin destabilization and demyelination. However, the exact sequence of events leading to structural abnormalities in myelin and axons remains to be clarified.

## 3. Discussion

In this study, we describe the effects induced by patient-derived anti-NMDAR (GluN1) IgG1 in mouse myelinating spinal cord cultures, and demonstrate neurotoxic and demyelinating effects of GluN1 targeting in vitro. This finding is in accordance with previous histopathological and MRI examinations in patients with anti-NMDAR-associated encephalitis, demonstrating the development of neurodegeneration in spinal neurons [6], grey matter atrophy [34] and white matter damage [7,35,36,37]. Moreover, diaplacentally transferred anti-NMDAR AABs cause severe neurotoxic effects on neonatal development [27] that may result in long-lasting neuropathological effects [38], corroborating our finding using in vivo models. 

Examination of neurite blebbing and fragmentation, which are features of neurodegeneration, revealed a higher percentage of lesioned neurons induced by SSM5 application in late myelinated cultures (DIV 28–42). This is the period (approximately DIV25 ± 5 days) when most myelinating oligodendrocytes formed mature myelin sheathes, as demonstrated in Figure 1 and previous reports [17,18]. Cultures treated in the early stages of the myelinating process (DIV 21–28) remained largely unaffected, indicating that susceptibility to SSM5 toxicity may likely be related to sheath maturation and compaction of the oligodendrocyte membrane around axons. In line with our previous data demonstrating the negative effects of SSM5 on NMDA-mediated Ca^2+^ responses and glucose transporter GLUT1 plasma membrane translocation in mature oligodendrocytes [15], we observed SSM5 immunoreactivity on the myelin compartment. Oligodendroglial NMDARs mediate spontaneous synaptic currents [39] and function as typical receptor-gated cation channels. Their stimulation can result locally from glutamatergic axons, with which oligodendrocyte form direct synapses [40,41]. Alterations in myelin NMDAR signaling and glucose uptake induced by SSM5 correlate with structural degenerative changes in neurites and myelin loss observed in the present study. These findings may hint at impaired glutamate-dependent axon-myelin communication (Figure 6). Our findings are in line with the proposed role of myelin NMDAR signaling in glycolytic support of myelinated axons that face energetic challenges when being physically isolated from the trophic metabolites (e.g., glucose, lactate) in the extracellular space [16,42,43]. The neurosupportive role of myelin NMDAR signaling was demonstrated in mutant mice, in which genetic ablation of the GluN1 subunit in cells of oligodendrocyte lineage underlies neurological deficits (hind limb clasping, hunchback, and ataxia). Particularly, GluN1 deficiency leads to ongoing axonal degeneration coinciding with myelin abnormalities (myelin delamination) in brain and spinal cord white matter [16]. With a greater similarity, we observe these processes in long-term myelinated co-cultures exposed to SSM5 (Figure 5). More recently, an altered expression of myelin NMDAR subunits has been linked to destabilization occurring in axon-myelin units in normal-appearing white matter of patients with multiple sclerosis [33]. Aberrant NMDAR signaling in axon-myelin synapses has been considered as a potential cause for the local detachment of myelin from axon, myelin blister formation that consecutively leads to axonal swelling, fragmentation, and demyelination [33]. In this regard, it is worth mentioning that a demyelinating disease can manifest along with anti-NMDAR encephalitis [7,44,45] and may share the hallmarks of classic multiple sclerosis lesions without complement deposition [46]. Accumulating clinical evidence indicate the presence of anti-NMDAR AABs in some patients with multiple sclerosis [47,48]. A recent study involving 200 patients revealed an overlap between anti-NMDAR encephalitis and MS (the German Network for Research on Autoimmune Encephalitis (GENERATE); in preparation). Nevertheless, it remains largely unclear whether antibodies against NMDARs or other neuronal surface structures [49] are causative for axonal pathology and demyelination in vivo.

However, axonal injury following prolonged treatment with SSM5 might be initiated by a direct impairment of the NMDAR transmission in spinal cord neurons. Internalization of NMDARs following crosslinking with anti-GluN1 antibody decreases synaptic cluster density of the receptor and affects glutamate signaling [9,10,11]. Several lines of evidence indicate that NMDAR hypofunction (NRHypo) may play a detrimental role in neuronal survival and may lead to apoptotic neurodegeneration. Chronic low neuronal activity induced in cultivated neurons by tetrodotoxin or blockade of basal extra-synaptic NMDAR activity with AP5 antagonist initiates expression of pro-apoptotic genes that leads to increased susceptibility to neuronal damage [50,51,52] (Figure 6). Synaptic NMDAR activity is coupled with the expression and recycling of several antioxidant enzymes [53,54] (Figure 6), while NRHypo may lead to neuronal injury upon oxidative damage [55,56]. Consistently, genetic ablation of GluN1 from cortical GABAergic neurons initiated increased formation of reactive oxygen species in cortical neurons [57]. Particularly relevant for the naturally occurring neurodegenerative process in long-term cultures [58], it was shown that synaptic NMDAR signaling activates µ-calpain and protects cortical neurons against starvation and oxidative stress-induced damage [59]. Transient blockade of NMDAR activity during early stages of development leads to widespread neurodegeneration and neuronal apoptosis [60]. Alternatively, NRHypo may lead to neuronal damage by altering the neural network circuit [12,61,62]. 

Effector mechanism of AABs can also be attributed to the antibody-dependent cellular cytotoxicity (ADCC) [63,64,65]. ADCC is a complement-independent mechanism involving Fc-receptor (FcR) activation by target-bound antibodies on the surface of effector cells, such as microglia. Microglial cells invade the spinal cord parenchyma at E11.5 [66] and take part in the early development of neuronal network, including the activity-dependent refinement of myelination via myelin sheath phagocytosis [67]. Myelinating co-cultures derived from the embryonic spinal cord comprise microglia/macrophages that efficiently recapitulate innate CNS immune properties [19]. The engagement of Fc-receptor (FcR) signaling on microglia by antigen-bound immunoglobulins triggers inflammatory [68] and proliferative [69] responses in the CNS. In cultured myeloid cells, Fc-driven reaction initiates innate effector responses including phagocytosis, inflammatory cytokine release and ADCC independent of complement activation [70,71,72,73]. Therefore, it would be of interest to determine the potential effect of anti-GluN1 on activation of microglia.

Neural cells are able to produce virtually all components of a complement system [74,75] that upon activation, may direct AABs’ pathogenicity via complement-dependent cytotoxicity (CDC). Indeed, complement activation is recognized as a major pathogenic or contributing factor in neuromyelitis optica spectrum disorders (NMOSDs) and myasthenia gravis (MG), where AABs form stable surface complexes on aquaporin-4 (AQP4) or acetylcholine receptors (AChRs), respectively [76,77,78]. In contrast, engagement of NMDAR upon cross-linking with anti-GluN1 lead to rapid internalization and receptor depletion from the cell surface, making it unlikely the antibodies will form IgG clusters and triggering subsequent complement activation. In support of this, histological examinations in anti-NMDAR encephalitis demonstrated prominent microgliosis but showed no evidences of complement deposits or complement-mediated tissue injury [79,80].

Our data expand the list of encephalitis-derived autoantibodies that mediate neurodegenerative changes in cultured neurons such as anti-GluR3B IgG [81] and anti-IgLON5 [31,32], and support the causative role of autoantibodies in autoimmune encephalitis. The results reported herein are of particular relevance for neurodevelopmental alterations caused by maternal-to-fetal transfer of NMDAR antibodies [82]. We propose that anti-GluN1 IgG1-induced neurotoxicity is associated with NMDAR hypofunction and disturbance of axon-myelin communication. The results of this study are relevant for other CNS disorders with similar neuropsychiatric manifestations, where NRHypo state is a critical component of the pathophysiology, such as schizophrenia [56,83], autism spectrum disorder [84], and Alzheimer’s disease [11,85,86].

## 4. Materials and Methods

### 4.1. Generation of Anti-NMDAR/GluN1 IgG1 Antibody

The generation and functional characterization of a recombinant IgG1 monoclonal antibody (SSM5) derived from intrathecal plasma cells of a patient with anti-NMDAR encephalitis and a control isotype against an irrelevant Cancer/Testis (CT) antigen, NY-ESO-1 (12D7) have previously been reported [14]. The effect of the biological activity of SSM5 and 12D7 on NMDA-induced Ca^2+^ responses in oligodendrocytes have previously been described [15].

### 4.2. Animals

All experimental procedures were conducted following the guidelines and protocols approved by the local animal welfare committee (Landesamt für Natur, Umwelt und Verbraucherschutz Nordrhein-Westfalen [LANUV]) under protocol number O74/08. Mice on C57BL/6 background were used for preparation of cell cultures.

### 4.3. Myelinating Cell Cultures

The method of generating myelinated spinal cord neurons was essentially based on previous publications [17], with minor modifications. Briefly, dissection and meninges stripping of mouse spinal cords (12.5 day of gestation) was performed in ice-cold Dulbecco’s Modified Eagle’s Medium (DMEM; 1000 mg/L glucose, 11880028). Six cords were combined in 1 mL of Hank’s Balanced Salts solution (HBSS; without Ca^2+^/Mg^2+^, 14175053) and digested with 0.25% trypsin (15090046) and 0.1% collagenase (C0130, Merck, Darmstadt, Germany) solution for 15 min at 37 °C. The digestion was stopped using 1 mL of SD solution [17] supplemented with 0.04 mg/mL DNase I (260913, Merck, Darmstadt, Germany). The cells were triturated in plating media (50% DMEM, 4500 mg/L glucose, 11960085; 25% heat-inactivated horse serum, 26050088; 25% HBSS, 14025050; 100 U/mL penicillin-streptomycin, 15140122) and plated at 150,000 cells per 100 μL on 96-multiwell dishes (165305, Greiner GmbH, Frickenhausen, Germany) or on coverslips (3 × 13 mm diameter, KHX4.1, Carl Roth GmbH, Karlsruhe, Germany) in 35 mm Petri dish (153066) pre-coated with 0.013 mg/mL poly-L-lysine solution (P4707, Merck, Darmstadt, Germany). The next day, 100 μL of differentiation media [17] was added to each well. Cells were grown at 37 °C in 5% CO_2_ and fed every second day by replacing half the volume with differentiation media. For day in vitro (DIV) 0–DIV14 media was supplemented with 10 µg/mL insulin (11061–68-0, Merck, Darmstadt, Germany). All reagents unless otherwise specified were purchased from Thermo Fisher Scientific (Meerbusch, Germany).

### 4.4. Treatment of Cells Grown on Multiwell Plates

To minimize ‘edge-effects’ due to the increased rate of evaporation or warming of media, wells on the outer edges of the plate were filled with HBSS, while cells were grown in the wells of the middle part of the plate. The effect of NMDA (100 µM; solubilized in water, Tocris Bioscience, Cat. No. 0114), dimethylsulfoxid (DMSO, 0.01% *v*/*v*, Sigma-Aldrich, D8418), triiodothyronine (T3, 1 µM, solubilized in DMSO, Sigma-Aldrich, T6397), SSM5 (10 µg/mL) and 12D7 (10 µg/mL) in myelinating cultures was investigated between DIV 21–28, DIV 28–35 and DIV 35–42. In order to avoid potential effects related to the location of the wells on the microplates, substances were added to the wells following a pre-generated random pattern that varied from one experimental repeat to the next. Four technical replicates per condition were used for each of four independent culture preparations. Treatments were performed three times a week by replacing half the medium. 

### 4.5. Antibodies and Immunocytochemistry

In general, cell cultures were fixed with 4% formaldehyde (ROTI^®^Histofix, P087.1, Carl Roth) for 15 min at room temperature (RT, 20 °C), washed in PBS, permeabilized in 0.5% Triton^TM^ X-100 (T8787, Merck, Darmstadt, Germany) in PBS for 30 min at RT, washed with PBS, and blocked with 10% goat (566380, Merck, Darmstadt, Germany) or donkey (566460, Merck, Darmstadt, Germany) serum in PBS (depending on the host of the secondary antibody) for one hour at RT. Primary antibodies were diluted in the blocking buffer and incubations were performed at RT for one hour or overnight at 4 °C. The following antibody were used: mouse anti-MBP (Merck Millipore, MAB381; 1:500), mouse anti-NF200 (Sigma-Aldrich, N5389; 1:500), rabbit anti-NF200 (Sigma-Aldrich, N4142; 1:500), rabbit anti-NG2 (Merck Millipore, AB5320, 1:400), human SSM5 IgG (10 µg/mL) and human 12D7 IgG (10 µg/mL). Bound antibodies were visualized after one hour incubation at RT by using appropriate combinations of species/isotype-specific fluorochrome-conjugated secondary antibodies: goat anti-mouse IgG (Alexa Fluor 647; Abcam 1:500), goat anti-rabbit IgG (Cy3, Merck Millipore, 1:500), donkey anti-human IgG (Alexa Fluor 647; Jackson ImmunoResearch, Cambridgeshire, UK, 709–605-149; 1:500), donkey anti-rabbit IgG (Alexa Fluor 488, Jackson ImmunoResearch, Cambridgeshire, UK, 711–545-152; 1:500), donkey anti-mouse IgG (Cy3, Jackson ImmunoResearch, Cambridgeshire, UK, 715–165-150; 1:500). Nuclei were counterstained with Hoechst 33258 (Thermo Fisher Scientific, Waltham, MA, USA; H3559; 2 µg/mL) for 5 min. Immunostained cultures were subsequently stored in PBS at 4 °C for imaging. Coverslips were mounted on glass slides in Immu-mount medium (Thermo Fisher Scientific, Waltham, MA, USA, 10622689).

### 4.6. Image Acquisition and Analysis

The Operetta CLS High Content Analysis System (PerkinElmer, Waltham, MA, USA) was used to acquire multiplexed wide-field fluorescent images in 96-well plate format with a 20 X objective, with the following channels 490/20 nm and 525/36 nm (Cy2, Alexa Fluor 488), 579/24 nm and 624/40 nm (Cy3) and 350/50 nm and 455/50 nm (Hoechst 33258, Thermo Fisher Scientific, Waltham, MA, USA). Images were captured from 25 fields of view per well and myelin (area stained with anti-MBP) and neurites (area stained with anti-NF200) were quantified using CellProfiler software [87,88] version 2.1.0. The pipelines used are based on dapi.cp and myelin.cp (last indexed on 24 March 2021) available at https://github.com/muecs/cp. Briefly, images were coded with parental metadata (imageID, wellID) and with row/column metadata. Quality control of the images was based on cell numbers (area stained with Hoechst) and artefacts. Shape was used to segment a closely-spaced cell. Total area, neurite area and myelin area measurements were exported to a comma-delimited spreadsheet for data analysis. The average value obtained from all images per well from each of the treatment conditions, from a single independent cell culture, were considered as one independent experimental unit.

For analysis of the neuronal injury and degeneration, the number of neurites that contained multiple or localized swelling (blebbing) or fragmentation of the neuronal process was counted manually using Adobe Photoshop software and expressed as a percentage of all neurites per automated optical inspection (AOI). The experimenter was blinded to the treatment condition during cell quantification.

Coverslips were imaged using an Olympus BX51 fluorescence microscope (Olympus France SAS, Rungis, France) and monitored by Cell^A imaging software (Soft Imaging System GmbH, Münster, Germany) and ImageJ, a free public-domain software developed by the National Institutes of Health (https://imagej.nih.gov). Detection of SSM5 and 12D7 immunoreactivity was performed on images acquired using a Leica SP8 confocal laser-scanning microscope (Leica, Wetzlar, Germany) and analyzed with LAS X software (Leica Application Suit X analysis tools (v. 3.7.4.23463).). Images were contrast-enhanced using Adobe Photoshop software (version CS3, Adobe Systems Ltd., Europe) to facilitate visibility in composite figures.

### 4.7. Statistical Analysis

Statistical analysis was performed using Prism software (GraphPad Software, San Diego, CA, USA). Statistically significant differences in myelination between multiple time points during culture cultivation and multiple treatments groups were determined by using one-way analysis of variance (ANOVA) followed by Tukey’s Honest Significant Difference (HSD) post hoc test. Data are represented as mean ± standard error of mean (SEM). In all experiments a *p* value: * < 0.05; ** < 0.01; *** < 0.001 was defined as statistically significant. Each *n*-value reported are derived from a single independent cell culture, comprising multiple wells or coverslips, representing technical replicates of the various treatments.

## Figures and Tables

**Figure 1 ijms-24-00248-f001:**
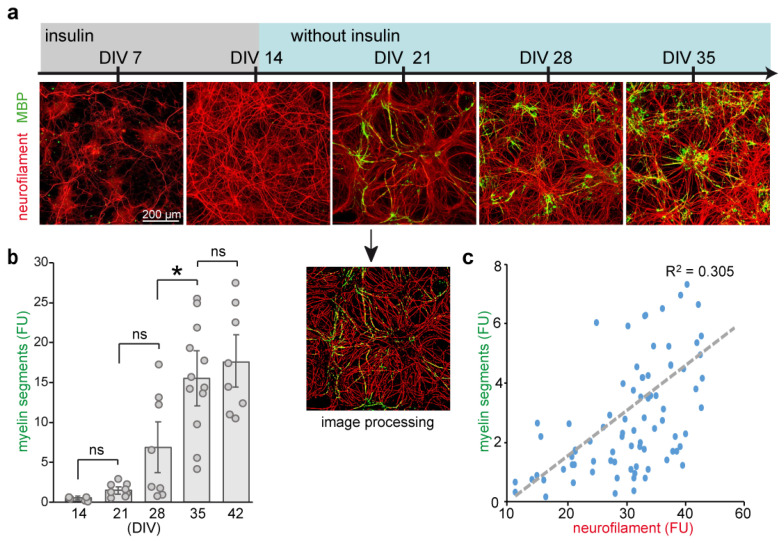
Characterization of myelinating culture derived from E12.5 mouse spinal cord. (**a**) Dissociated cells from embryonic spinal cord were grown on poly-L-lysine coated 96-well plates supplemented with insulin (up to DIV 14). Progression of myelination over 5 weeks in culture was identified using antibody to intermediate neurofilament heavy chain (NF200; red) and MBP (green), the lineage markers specific for neurons and mature myelinating oligodendrocytes, respectively. (**b**) Quantification of myelination. Overlap MBP signal with NF200 enable to visualize myelin segments. Double immunostained images were processed with a CellProfiler pipeline (example shown for DIV 21) that enables quantification of the pixel area for each signal. Myelin segments were calculated as an overlay of red and green lines. (**c**) At the peak of myelination (DIV 35), the extent of neuronal insulation is proportional to neurite density in individual wells. Data summarized results from three independent experiments (*n* = 8). Mean ± SEM. ANOVA followed by post hoc Tukey’s analysis revealed a significant increase in myelination in DIV 28 vs. DIV 21, DIV 35 vs. DIV 28, but not in the late stages (DIV 42 vs. DIV 35). * *p* < 0.05; ns, non-significant Pearson’s correlation was calculated as R^2^ = 0.305. DIV, days in vitro. FU, fluorescent units. Scale bar: 200 µm.

**Figure 2 ijms-24-00248-f002:**
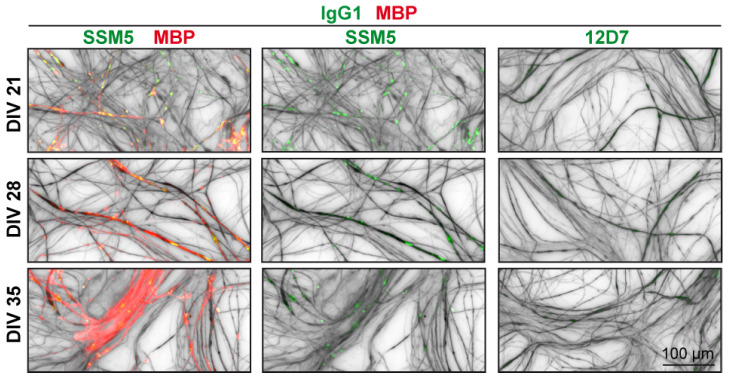
Immunofluorescence analysis of the reactivity of SSM5 in myelinating spinal cord cultures. SSM5 (green) antibodies target neurites, immunostained for NF200 (grey) and myelin structures (MBP, red). Note: dot-like immunostaining of neurites with SSM5 was observed prior to myelination (DIV21). Lack of immunoreactivity of non-CNS reactive isotype control (12D7). Scale bar: 100 µm.

**Figure 3 ijms-24-00248-f003:**
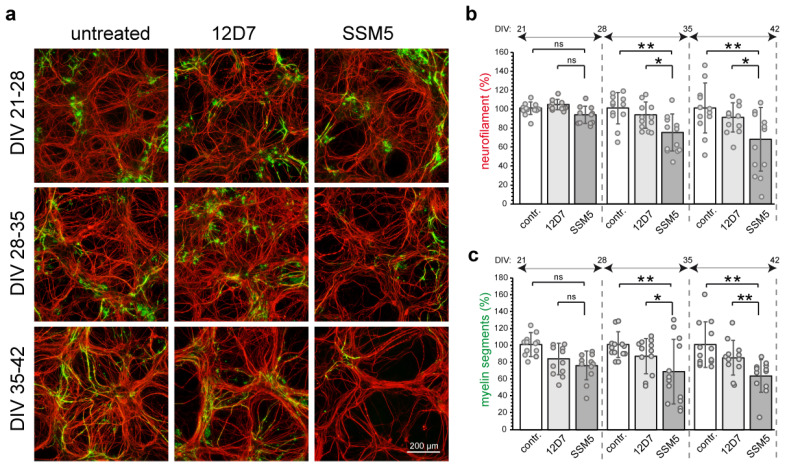
SSM5 antibodies have drastic effects on myelin and neuronal density. (**a**) Immunofluorescence analysis of neurites (NF200; red) and myelin (MBP; green) after treatment with SSM5 (10 µg/mL) or 12D7 (10 µg/mL) for one week at different stages of myelination. Scale bar: 200 µm. Quantification of neurofilament density (**b**) and myelin segments (**c**) following treatment with SSM5 starting at DIV 21, DIV 28, DIV 35. One-way ANOVA with Tukey multiple comparison, showed significant differences between SSM5 vs. 12D7 vs. control groups; *n* = 4; technical replicates = 3; * *p* < 0.05, ** *p* < 0.01; ns, non-significant. Note: detrimental effect of SSM5 was more pronounced at later stages of culture and reached statistical significance after DIV 28. Mean ± SEM.

**Figure 4 ijms-24-00248-f004:**
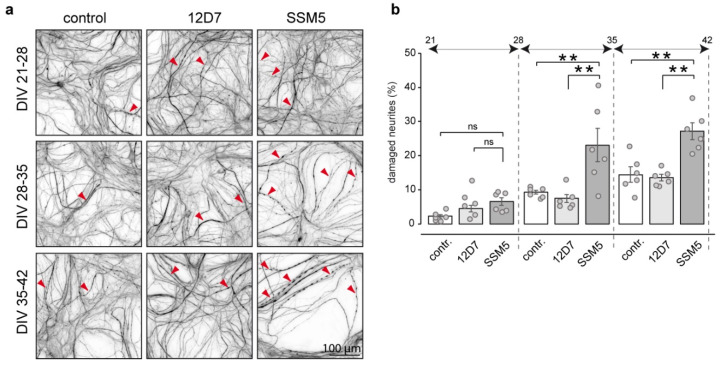
Treatment with SSM5 affects neurite integrity and leads to neurite swelling and fragmentation. (**a**) Sample images of non-treated control cultures (left panels), vs. cultures treated for one week with 12D7 (middle panels) or SSM5 (right panels) and immunostained for NF200 (grey). Red arrowheads indicate affected neuronal processes. Scale bar: 100 µm. (**b**) Quantification of damaged neurites. Data show the percentage of processes exhibiting morphological abnormalities (swelling, fragmentation) of total number of neurites. Three independent experiments (technical replicates = 2), Mean ± SEM. One-way ANOVA with Tukey multiple comparison showed significant differences between SSM5 vs. 12D7 vs. control groups only when treatment was performed in late cultures (DIV 28–42); ns, non-significant; ** *p* < 0.01.

**Figure 5 ijms-24-00248-f005:**
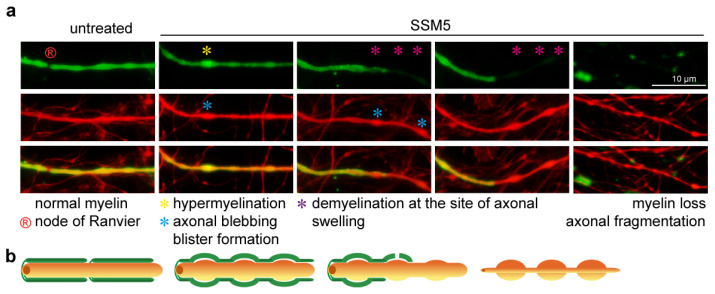
Treatment with SSM5 leads to axonal degeneration and demyelination. (**a**) Representative images of non-treated control (left panels), vs. cultures treated for one week with SSM5 (middle and right panels) and immunostained for NF200 (red) and MBP (green). Asterisks mark hyperintense myelin (yellow), axonal swelling (blue), demyelinated area (purple) on affected axons. Node of Ranvier marked with ^®^. Scale bar: 10 µm. (**b**) schematic representation of the different types of swelling analyzed on myelinated axons (corresponding to image panels above). Proposed sequence of events: normal myelinated axon with node of Ranvier (left panel), formation of axonal bleb coincides with myelin hyperintensity that may refer to a blister-like structure (middle panels). Deterioration of myelin sheath occurs at the site of axonal injury, while axonal blebbing is further increased on demyelinating axons (right panel).

**Figure 6 ijms-24-00248-f006:**
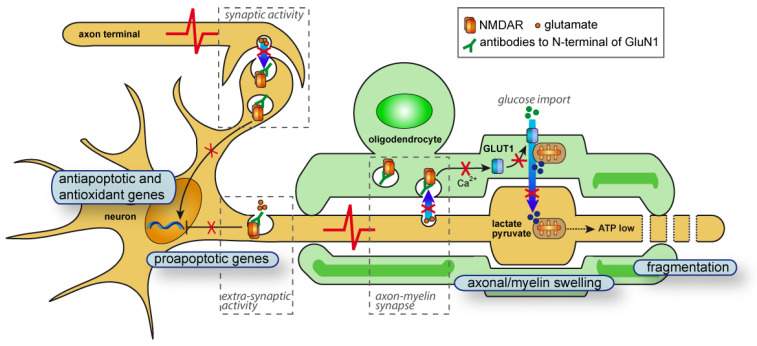
Graphic representation of the putative mechanisms involved in neurotoxic effects of human anti-GluN1 autoantibodies on myelinated neurons. Anti-GluN1 IgG1 may interfere with neurotransmission through NMDARs localized at different parts of neuron. Targeting NMDAR at axon-myelin synapses will affect glutamate-mediated localization of glucose transporter (GLUT1) to surface membrane of oligodendrocyte and diminish trophic support of the axon. Subsequently, the insufficient local ATP production by axonal mitochondria may trigger neurodegenerative processes at internodes. Anti-GluN1-mediated surface depletion of extra-synaptic NMDARs may affect inhibitory effects of basal NMDAR activity on expression of proapoptotic genes, such as *Apaf1*, *Bid*, *Puma*. Lowering synaptic NMDAR signaling may interfere with its known role in CREB-mediated neuronal survival, neuroprotective µ-calpain activation, antiapoptotic and antioxidant gene expression.

## Data Availability

Not applicable.

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
