# Peer review of "Myelinating Co-Culture as a Model to Study Anti-NMDAR Neurotoxicity"

_ijms, 2022, doi:10.3390/ijms24010248_

Round 1
Reviewer 1 Report
The submitted study demonstrates the effects of patient-derived GluN1-specific monoclonal antibody on the development of neuronal networks and oligodendrocytes maturation in vitro. The study is interesting and can be recommended for the publication in IJMS, but some comments have to be addressed.
1. The authors clearly demonstrate effects of the antibodies on neurons and oligodendrocytes, but ignore astrocytes and microglial cells. Abstract contains the next phrase: "spinal cord cell culture model that contains all major CNS cell types". Could the authors provide the precise percentage of cell populations in their cultures at different stages of the cultivation? I did not find this information in the previously published work (reference 17). This question is pivotal since other types of glial cells express NMDARs that are being targets for anti-NMDAR antibodies and NMDAR agonists. If astrocytes are present in cultures, please, describe their phenotype (reactive/non reactive). How do the NMDA, T3 and insulin exposures affect the percentages?
2. The authors use the terms "neural" and "neuronal" in the text. Are any differences in the meaning of these terms?
3. Is NF200 a marker of mature neurons? Please indicate it clearly in the text.
4. The authors demonstrate effects of the antibodies on the morphology of cells (neurite outgrowth, myelinization etc.) and state that neurodegeneration and neuronal injury occur. It would be interesting to see at least the percentage of live/dead cells evaluated with Hoechst and propidium iodide or 7AAD for example. It is even better, if the authors will evaluate the loss of exactly neurons.
5. What was the glycine concentration in the experiments?
6. Please provide a rationale using paired t-test instead of unpaired test in Fig. 1.
Author Response
1. The authors clearly demonstrate effects of the antibodies on neurons and oligodendrocytes, but ignore astrocytes and microglial cells. Abstract contains the next phrase: "spinal cord cell culture model that contains all major CNS cell types". Could the authors provide the precise percentage of cell populations in their cultures at different stages of the cultivation? I did not find this information in the previously published work (reference 17). This question is pivotal since other types of glial cells express NMDARs that are being targets for anti-NMDAR antibodies and NMDAR agonists. If astrocytes are present in cultures, please, describe their phenotype (reactive/non reactive). How do the NMDA, T3 and insulin exposures affect the percentages?
We thank the reviewer for raising this issue and consider the potential involvement of glial cells in anti-GluN1 pathogenicity (Discussion part, page 7). Unfortunately, the examination of glial cell morphology and composition lied outside the scope of this study. Taking into account the importance of such characterization, we provide the additional references for publications describing the composition of glial cells in primary cultures of mouse E12.5-E13.5 spinal cord (page 2). In brief, while the neurons and oligodendrocytes are the major cell types (e.g. 38% of neurons and 28% of oligodendrocytes at DIV 10 (Pang et al. Brain Behav. 2012, PMID: 22574274), GFAP+ astrocytes and Iba1+ (or CD11b+) microglia are also present at all stages examined (Bijland et al. F1000Res. 2019, PMID: 31069065) (Vargova et al 2022, PMID: 35197829). At DIV 10 typical culture contains 10% of astrocytes, and 10% of microglia/macrophage (Pang et al. Brain Behav. 2012, PMID: 22574274) and their percentage is increased over time (for example up to 16% of astrocytes at DIV 14 (Mikhailova et al. Int J Neurosci. 2019, PMID: 30621485)).
2. The authors use the terms "neural" and "neuronal" in the text. Are any differences in the meaning of these terms?
Following to the traditional acceptance and definition of both terms in literature, we use the term “neural” to refer to any type of cells in the nervous system including neurons, astrocytes, oligodendrocytes and microglia; whereas the term "neuronal" is specifically related to neurons.
3. Is NF200 a marker of mature neurons? Please indicate it clearly in the text.
We include the references and statement indicating NF200 as a marker of neuronal differentiation in the text (page 3).
4. The authors demonstrate effects of the antibodies on the morphology of cells (neurite outgrowth, myelinization etc.) and state that neurodegeneration and neuronal injury occur. It would be interesting to see at least the percentage of live/dead cells evaluated with Hoechst and propidium iodide or 7AAD for example. It is even better, if the authors will evaluate the loss of exactly neurons.
Regarding this point, in our original manuscript we demonstrate the results of injured neurites that exhibit signs of neurodegeneration (blebbing or fragmentation) (Figure 4) and loss of neurons according to immunofluorescent analysis with anti-NF200 antibodies (Figure 3b). To provide further insights into the mechanism and type of cell death it would require examination that is more detailed. In general, neurodegenerative processes in neurites are hardly correlated with analysis of cell death (Jellinger and Stadelmann, 2001, PMID: 12214070) (Andreone et al. 2020, PMID: 31451511) or even can be uncoupled with neuronal death (Ikegami and Koike, 2003, PMID: 14622905) (Fukui, 2016. PMID: 27895381). Nevertheless, we made an effort to quantify the percentage of abnormal nuclei that exhibit signs of pyknosis or karyorrhexis. Our data are consistent with previous observation, indicating cytotoxic effect of the SSM5 antibodies (Figure 3-4). The data are included in a new supplementary Figure S3 and presented on page 5.
5. What was the glycine concentration in the experiments?
We included the estimated concentration of glycine in the main text (page 3). Considering the culture medium composition the free glycine concentration can be calculated as 0.2-0.25 mM (50% DMEM (0.4 mM of glycine) + 25% Horse serum (0.1-0.2 mM of glycine (Bergero et al, J Anim Physiol Anim Nutr (Berl). 2005. PMID: 15787986; Westfall et al., J Natl Cancer Inst. 1954. PMID: 13233865)) + 25% HBSS (0 mM of glycine).
6. Please provide a rationale using paired t-test instead of unpaired test in Fig. 1.
We thank the reviewer for pointing to this mistake. We have re-calculated row data by using analysis of variance (ANOVA) followed by post hoc Tukey’s analysis and included changes in the Figure 1 (page 3) and “Materials and Methods” part (statistical analysis; page 10).
Reviewer 2 Report
In this manuscript, the authors addressed Anti-GluN1 AABs induce neurodegenerative changes and associated myelin loss in myelinated spinal cord cultures.The emerging association of NMDAR encephalitis with demyelinating disorders is an interesting and important question, but this paper doesn’t provide enough evidence that higher vulnerability of myelinated neurons to Anti-GluN1 AABs. The neurodegenerative happens at late stage of culture is not necessarily related to the increased myelination. At this stage, the neuron itself or other cell types (e.g., Microglia in the mixed culture) can also be the reason for the more toxic effect. More suitable control experiments are neeed for this conclusion.Another question is that the myelin defects after adding SSM5 can be caused by OL cells itself or by axon loss as a secondary effect. To decipher if SSM5 directly induces myelin loss or not will make this paper more meaningful. In general, this paper needs clearer evidence to show how Anti-GluN1 AABs affect neurons, oligodendrocytes, and their glutamatergic communication.
Author Response
In this manuscript, the authors addressed Anti-GluN1 AABs induce neurodegenerative changes and associated myelin loss in myelinated spinal cord cultures. The emerging association of NMDAR encephalitis with demyelinating disorders is an interesting and important question, but this paper doesn’t provide enough evidence that higher vulnerability of myelinated neurons to Anti-GluN1 AABs. The neurodegenerative happens at late stage of culture is not necessarily related to the increased myelination. At this stage, the neuron itself or other cell types (e.g., Microglia in the mixed culture) can also be the reason for the more toxic effect. More suitable control experiments are need for this conclusion.
We thank the reviewer for this comment and understand that further experiments are required to dissect the mechanism involved. Indeed, similar to long-term non-myelinated neuronal cultures (as described elsewhere), we observed gradually increased number of injured neurites over time (Figure 4b), strengthening the role of active (calpain-dependent) neurodegenerative-like neuronal death in long-term cultures (Kim et al. (Exp.Mol.Med 2007, PMID: 17334225). In this regard, neuronal deterioration by SSM5 might also be relevant for neuroprotective role of synaptic (vs extra-synaptic) NMDAR signalling via activation of μ-calpain (Wang et al., J Neurosci, 2013, PMID: 24285894). Discussion of this aspect of neuronal vulnerability in long-term cultivation is now included in page 7 and the legend of Figure 6.
We completely agree with the reviewer and consider the different scenarios of SSM5 toxicity, some of them had been described in discussion part. We corrected the text in “Abstract” part (page 1): These findings may point to the higher vulnerability of myelinated neurons towards interference in glutamatergic communication, and may refer to the understanding of the emerging association of NMDAR encephalitis with demyelinating disorders.
Another question is that the myelin defects after adding SSM5 can be caused by OL cells itself or by axon loss as a secondary effect. To decipher if SSM5 directly induces myelin loss or not will make this paper more meaningful. In general, this paper needs clearer evidence to show how Anti-GluN1 AABs affect neurons, oligodendrocytes, and their glutamatergic communication.
To clarify this issue we would need to establish cultures from conditional knockout mice to delete GluN1 specifically in oligodendrocytes. Currently, these mice are not housed in our facility; though, we consider the importance to use these models for our ongoing investigation. Another possibility would be analysis of ultrastructural damage of myelin following anti-GluN1 ligation, such as myelin delamination. We thank the reviewer for raising this central issue; however, at present we are not able to provide an exact answer.
Following reviewer’s suggestion, we would like to supplement the manuscript with microscopy analysis we performed (new Figure 5a), and our current vision of complex interplay between demyelination and axonal degeneration in this model (new Figure 5b, text on page 6 and page 8). Previous study demonstrated a co-existence of axonal degeneration and myelin abnormalities (myelin delamination) in spinal cord white matter tracts of conditionally GluN1-deficient mice (Figure 6E in Saab et al, PMID: 27292539). Hence, ablation of NMDAR signalling in oligodendrocytes ultimately leads to myelin deterioration in parallel to axonal degeneration. With a greater similarity, we observe these processes in myelinated co-cultures exposed to SSM5 (new Figure 5). The earliest pathological event we noticed is hyperintensity of myelin sheath (that resembles the myelin blister-like formation; see Luchicchi et al., Annal Neurol, 2021, PMID: 33410190) at the site of axonal bleb (new Figure 5). Remarkably, “blistering” is apparently followed by deterioration of myelin thickness (demyelination), while axonal blebbing become more pronounced on demyelinating axons. According to our present data, myelin abnormalities coincide with axonal degeneration; however, the exact sequence of events remains to be clarified.
Round 2
Reviewer 1 Report
All my comments have been addressed.
Author Response
We thank reviewer for constructive questions
Reviewer 2 Report
The conclusions of the revised version are more reasonable and are supported by the results. One minor question--in the new Figure S3, authors show SSM5 antibiody increases cell death in late cultures. Is it possible to co-stain with different cell markers (neuron, oligodendrocyte, microglia etc.) to see what kind of cells are susceptible to the SSM5 antibiody?
Author Response
Following reviewer’s suggestion, we attempted to perform the analysis of neuronal and oligodendroglial cell death in response to the SSM5. In neurons undergoing degeneration (blebbing) we did not observed morphologically visible nuclear abnormalities (Figure 1, not included in the manuscript). Neuronal somata of dead cells was not co-stained with NF200. Some dead cells adjacent to severely injured (fragmentation) neurites are presumably neurons (Figure 1, marked with hashtag). However, co-staining with alternative neuronal marker is required to identify neuronal nature of these dead cells. Likewise, the corps of apoptotic cells were not labelled with anti-MBP antibody (Figure 2, marked with hashtag). Unfortunately, at present we are not able to elucidate a direct effect of SSM5 treatment on neuronal or oligodendroglial cell death.
Concerning oligodendrocytes, we observed an abnormal distribution of MBP toward soma of degenerating oligodendrocytes, which exhibit signs of hypomyelination, myelin blistering or association with myelin debris (Figure 2, not included in the manuscript). This is highly reminiscent of the pattern seen in oligodendrocytes with presumably affected mRNA transportation into processes (Silva et al., Nat Commun., 2019, PMID: 31371763; Lyons et al., Nat Genet, 2010, PMID: 19503091; Sen et al. Front Neurosci. 2021, PMID: 33841096). We use this parameter (accumulation of MBP in cell body) as a potential hallmark of oligodendrocyte degeneration to estimate the susceptibility of myelinating oligodendrocytes to the SSM5 antibody (Figure 2b). Our data are consistent with previous observation, indicating detrimental effect of the SSM5 on myelination (existing Figure 3). Nevertheless, oligodendrocytes with affected/abnormal myelin demonstrate healthy nuclear morphology (Figure 2a).
For details and Figures, see an attached .pdf file
